# Chaos Synchronization of Integrated Five-Section Semiconductor Lasers

**DOI:** 10.3390/e26050405

**Published:** 2024-05-06

**Authors:** Yuanyuan Guo, Yao Du, Hua Gao, Min Tan, Tong Zhao, Zhiwei Jia, Pengfa Chang, Longsheng Wang

**Affiliations:** 1Key Laboratory of Advanced Transducers and Intelligent Control System, Ministry of Education and Shanxi Province, Taiyuan 030024, China; oay_ud@163.com (Y.D.); 9966tzm@sina.com (M.T.); zhaotong@tyut.edu.cn (T.Z.); jiazhiwei@tyut.edu.cn (Z.J.); changpengfa@tyut.edu.cn (P.C.); wanglongsheng@tyut.edu.cn (L.W.); 2College of Electronic Information and Optical Engineering, Taiyuan University of Technology, Taiyuan 030024, China; 3College of Information Engineering, Shanxi Vocational University of Engineering Science and Technology, Taiyuan 030024, China; gaohua@sxgkd.edu.cn

**Keywords:** chaotic laser, chaos synchronization, chaotic communication, key space

## Abstract

We proposed and verified a scheme of chaos synchronization for integrated five-section semiconductor lasers with matching parameters. The simulation results demonstrated that the integrated five-section semiconductor laser could generate a chaotic signal within a large parameter range of the driving currents of five sections. Subsequently, chaos synchronization between two integrated five-section semiconductor lasers with matched parameters was realized by using a common noise signal as a driver. Moreover, it was found that the synchronization was sensitive to the current mismatch in all five sections, indicating that the driving currents of the five sections could be used as keys of chaotic optical communication. Therefore, this synchronization scheme provides a candidate to increase the dimension of key space and enhances the security of the system.

## 1. Introduction

Chaotic optical communication has attracted much attention because of its fast transmission rate [1,2,3,4], capacity for long-distance communication [5,6] and compatibility with current fiber-optic network communications [7,8,9]. For instance, in 2005, Argyris et al. realized a metropolitan area network trial with a rate of 1 Gb/s and a communication distance of 120 km in Athens [7]. In 2010, Argyris et al. realized 100 km, 2.5 Gb/s chaotic optical communication using a photonic integrated chaotic laser [10]. In 2018, Yi et al. realized 100 km, 30 Gb/s chaotic optical communication using chaotic optoelectronic oscillators [2]. External cavity feedback semiconductor lasers are the most commonly used transceivers in chaotic optical communications because of their simple structure and flexible operation [11]. Externally controllable hardware parameters such as bias current, feedback strength and external cavity feedback length of external cavity feedback semiconductor lasers can be used as a key for chaotic optical communication. However, the outer cavity feedback length can be obtained by autocorrelation or mutual information calculation of the timing waveforms. The relaxation oscillation frequency of the laser can be determined by analyzing the spectrum of the chaotic signal, and then the corresponding bias current can be estimated [12]. As a result, in this communication system, the number of hardware parameters that can be used as key parameters is smaller, the key space of chaotic optical communication is insufficient, and its security is threatened.

Therefore, in recent years, a large number of key space enhancement schemes to enhancing the key space of the chaotic optical communication have been proposed. For instance, in 2009, Xia et al. proposed that the time-delay feature can be suppressed by two external cavities [13]; thus, the cavity length can be used as a key parameter and the key space can be enhanced. In 2016, Yi et al. introduced 16 cascaded standard fixtures in an optoelectronic feedback loop to suppress the time-delay information and increased the key space of the system to 10^48^ by taking the center frequencies of the standard fixtures as additional key parameters [14]. In 2019, Wang et al. numerically demonstrated that external feedback-based chirped fiber Bragg grating (CFBG) can eliminate the time-delay feature, and the dispersion and center frequency of the CFBG were introduced as key parameters, increasing the key space to 2^44^ [15]. Jiang et al. enhanced key space by modulating the phase of the feedback light using a pseudo-random code sequence with a code length of 2^7^ [16]. Ji et al. proposed a scheme of secondary encryption to increase the dimensionality of key parameters, realizing the enhancement of the key space for chaotic communication [17].

In this paper, we propose a scheme of chaos synchronization between two integrated five-section semiconductor lasers (IFSSL) and model the simulation study in the commercial software program VPItransmissionMaker. The IFSSL is composed of two distributed feedback laser sections (DFB1 and DFB2), two phase sections (P1 and P2) and an amplification section (A). We investigate the effects of the driving currents of the five sections on the output dynamic characteristics and determine the parameter range to generate chaotic driving currents, and their mismatch on the synchronization coefficients is studied in detail, proving that the synchronization is sensitive to the mismatching of the driving currents. Predictably, in the chaotic optical communication system based on IFSSL, the driving currents of all five sections of IFSSL can be used as the key parameter, which increases the dimension of the key space and enhances the key space of the system.

## 2. Materials and Methods

Figure 1 shows the schematic diagram of the chaos synchronization of two IFSSLs. The broadband noise signal generated by the super-luminescent diode (SLD) is transmitted unidirectionally through an optical isolator (OI) and then filtered by a low-pass filter (LPF). Through a 3 dB optical coupler (OC), the broadband noise signal is split into two paths which are, respectively, injected into the IFSSLA and IFSSLB at the legitimate communication parties, Alice and Bob. The injection powers of IFSSLA and IFSSLB are varied by private variable optical attenuators (VOAs). The output of the IFSSL is filtered by a low-pass filter (LPF) and divided into two paths through the OC (optical coupler); one path is directly detected, optical spectra are captured by an optical spectrum analyzer (OSA), and the other path is converted into an electrical signal by the photodetector (PD), which views and captures the R-F spectrum, with an electrical spectrum analyzer (ESA), and the temporal waveform of the signal, with an oscilloscope (OSC). The IFSSL is composed of two distributed feedback laser sections (DFB1 and DFB2), two phase sections (P1 and P2) and an amplification section (A). The P1 section is used to change the coupling phase between DFB1 and DFB2; the P2 and A sections are used to generate active optical feedback, which the P2 section uses to change the feedback phase of the optical feedback, and the A section is used to provide gain and change the intensity of the feedback light. The five sections are driven by individual currents. Table 1 shows the main internal design parameters of the IFSSL.

## 3. Results

### 3.1. Dynamic Characterization of IFSSL

Figure 2 shows the temporal waveforms, optical spectra and radio frequency (R-F) spectra of the output signal of IFSSL under various *I*_P1_ values, when *I*_DFB1_ = 60 mA, *I*_DFB2_ = 66 mA and *I*_P2_ = *I*_A_ = 0 mA. As shown as Figure 2(a1–a3), when *I*_P1_ = 9 mA, the IFSSL output is a single-mode signal whose center wavelength matches the main mode of the DFB2 laser *λ*_2a_ (*λ*_2a_ = 1557.989 nm). The temporal waveform is almost a constant with small ripples, and the R-F spectrum exhibits oscillations which are similar to noise. Thus, the IFSSL is currently in a stable state. When *I*_P1_ increases to 12 mA, the IFSSL generate a dual-mode signal, indicating a period-one state, as shown in Figure 2(b2), in which a composite cavity mode [18,19,20] *λ*_2b_ appears close to *λ*_2a_. Furthermore, as shown in Figure 2(b1,b3), the corresponding temporal waveform shows a single-period oscillation trace, and the R-F spectrum displays significant peaks at the frequencies corresponding to the composite, *f*, and its higher harmonics, *f*. When *I*_P1_ further increases to 15 mA, the IFSSL shifts to a quasi-periodic state, as shown in Figure 2(b3); in addition, at the wavelengths *λ*_2a_ and *λ*_2b_, the wavelength of *λ*_1_ is lased. The temporal waveform in Figure 2(c1) exhibits multi-period oscillation, and the R-F spectrum in Figure 2(c3) presents multiple frequency components *f*, 2*f*, *f*_a_ (frequency detuning of *λ*_1_ and *λ*_2a_), *f*_b_ (frequency detuning of *λ*_1_ and *λ*_2b_) and *f* − *f*_a_. When *I*_P1_ = 17 mA, the IFSSL generates chaotic signal, as shown in Figure 2(d1–d3), the optical spectra and the R-F spectrum are broadened, and the temporal waveform shows large and irregular magnitude variations.

Figure 3 shows a diagram of the dynamic states of the IFSSL in the parameter space *I*_DFB2_ and *I*_P1_ when *I*_P2_ = *I*_A_ = 0 mA and *I*_DFB1_ = 60 mA. Different colors denote different dynamical states. By varying *I*_DFB2_ and *I*_P1_, the IFSSL can generating a variety of dynamical states, such as a steady state (S), a period-one state (P1), a quasi-periodic state (QP) and chaos (C). When 43 mA < *I*_DFB2_ < 100 mA, 0 mA < *I*_P1_ < 30 mA, as shown in Figure 3, the IFSSL is mainly in chaos (C) and a steady state (S). Moreover, a quasi-periodic (QP) state occurs less frequently, and a period-one (P1) state occurs in the smallest range. It can be observed from the map that the IFSSL is capable of outputting a chaotic signal over a wide parameter range of driving currents of *I*_DFB2_ and *I*_P1_.

In order to comprehensively analyze the effect of the driving current of the feedback loop, we select four points a, b, c, d in Figure 3 to study the output state of the IFSSL under different *I*_P2_ and *I*_A_ when the IFSSL in four different states. The diagram of the dynamic state is shown in Figure 4. When the IFSSL is initially at point a, the original steady state rapidly switches to a period-one state, a quasi-periodic state or a chaotic state as the *I*_P2_ and *I*_A_ vary. As shown as Figure 4a, the quasi-periodic state occupies the major portion, the chaotic state and the period-one state occupies the minor portion, and the steady state occupies the smallest portion. When the IFSSL is initially in a period-one state at point b, as shown in Figure 4b, the range of the chaotic state and period-one state portions expand and those of the steady state and quasi-periodic state portions narrow compared with Figure 4a. The steady state disappears in Figure 4c when the IFSSL is initially in a quasi-periodic state at point c. The period-one state occupies a small portion, and the quasi-periodic state and chaotic state still occupy a major portion, in which the range of the chaotic state further expands compared with Figure 4a,b. The initially chaotic state of IFSSL at point d changes to a quasi-periodic state and a period-one state in certain areas as the *I*_P2_ and *I*_A_ are adjusted. As the results in Figure 4d show, the range of the chaotic state is largest in Figure 4a–d, and the range of the quasi-periodic state is close to that of a chaotic state. In conclusion, the IFSSL can always generate a chaotic signal when the *I*_P2_, *I*_A_, *I*_DFB2_ and *I*_P1_ vary.

### 3.2. Chaos Synchronization of IFSSL

Chaos synchronization is prerequisite for chaotic secure communication. In this scheme, the chaos synchronization between two responses IFSSLA and IFSSLB with identical parameters is achieved using a broadband noise signal generated from the SLD as the common driving signal. Figure 5 shows the temporal waveforms of the SLD, the response IFSSLA and IFSSLB and the corresponding correlation plots of SLD and IFSSLA and between IFSSLA and IFSSLB. The power of SLD is 30.18 mW, and the currents of IFSSLA and IFSSLB both are set as *I*_DFB1_ = 60 mA, *I*_P1_ = 26 mA, *I*_DFB2_ = 77 mA, *I*_P2_ = 15 mA and *I*_A_ = 15 mA. Figure 5a shows that the chaotic temporal waveforms of IFSSLA and IFSSLB are mostly identical, the corresponding scatter plots in Figure 5b exhibit a linear distribution, and the correlation coefficient reaches 0.9742. Figure 5c shows the temporal waveforms of the SLD and IFSSLA, which have no obvious similarities. The scattered distribution in Figure 5d also proves that the correlation between SLD and IFSSL is very low, and the correlation coefficient is calculated as 0.3868. The high-quality synchronization between the two responses lasers benefits the low error decoding of message, while the low correlation between SLD and IFSSLA ensures the security of the communication system by preventing eavesdroppers from obtaining carrier-correlated informationvia a public link.

The effects of the power of SLD on the synchronization quality of IFSSLA and IFSSLB when the *I*_DFB1_, *I*_DFB2_ and *I*_A_ have different values are shown in Figure 6. For all three cases, as the driving power increases, the correlation coefficient of the IFSSLA and IFSSLB increases, and when the driving power is larger than about 30 mW, high-quality chaos synchronization (CC > 0.9) can be achieved. As the driving current *I*_DFB1_ in Figure 6a increases, the power coupled into the DFB2 region from the DFB1 region increases, and thus the driving power required to achieve high-quality synchronization slightly decreases. With the increase of the drive current *I*_DFB2_ and *I*_A_, as shown in Figure 6b,c, the output power of the laser DFB2 increases, and thus the drive power required to achieve high-quality chaos synchronization slightly increases.

On the basis of the high-quality chaos synchronization of IFSSLA and IFSSLB described above, we investigate the effects of the mismatching of the driving current *I*_DFB1_, *I*_DFB2_, *I*_P1_, *I*_P2_ and *I*_A_ on the synchronization coefficient of IFSSLA and IFSSLB. Figure 7 shows that the synchronization performance of the system becomes degradative as the current mismatch increases, but the effect of individual current mismatches on the synchronization performance is different. When the synchronization of the system decreases to the threshold value of 0.90, the mismatch ranges of *I*_DFB1_, *I*_P1_, *I*_DFB2_, *I*_P2_ and *I*_A_ are 0.297 mA, 0.334 mA, 0.193 mA, 0.122 mA and 0.226 mA, respectively. This indicates that the synchronization performance of the system is extremely sensitive to the current mismatch of all five sections of the IFSSL, resulting that the driving currents of the DFB1, P1, DFB2, P2 and A sections can all be used as key parameters to enhance the key space of the chaotic optical communication system.

## 4. Discussion

Common transceivers used for chaotic optical communication systems often exhibit some problems, such as low integration and hardware parameter which can be used as key parameter is few. Leading to insufficient key space for chaotic optical communication. In this work, we propose an integrated five-section semiconductor laser IFSSL and achieve chaos synchronization between two IFSSLs by common driving with a SLD. Through simulation studies, we demonstrate that the driving currents of all five sections of our proposed IFSSL can be used as the key parameters of the chaotic optical communication system, promising a huge additional key space. We expect that this study will provide a new idea to enhance the key space of chaotic optical communication systems.

## 5. Conclusions

In conclusion, we achieved chaos synchronization between two integrated five-section semiconductor lasers IFSSLA and IFSSLB, which has promise for use as a transceiver to enhance the key space of chaotic optical communication. We built the chaos synchronization system using a SLD as the driver in VPIcomponentMaker simulation software. First, we studied the dynamic characteristics of the IFSSL and found the path to chaotic state (steady state—period-one—quasi-periodic—chaotic state). The results also indicated that the chaos could be generated from an IFSSL within a large range of driving current parameters for the five sections. Subsequently, the effects of the driving power and the mismatch between the driving currents of the five sections on the chaos synchronization coefficients of two IFSSLs are researched in detail. The results demonstrated the sensitivity of the synchronization coefficient to the driving current mismatch of the five sections. Therefore, the driving currents of the five sections of the IFSSL can be used as the key parameters for chaotic optical communication systems to improve the dimension of the system key space, which will be one of the focuses of future research work.

## Figures and Tables

**Figure 1 entropy-26-00405-f001:**
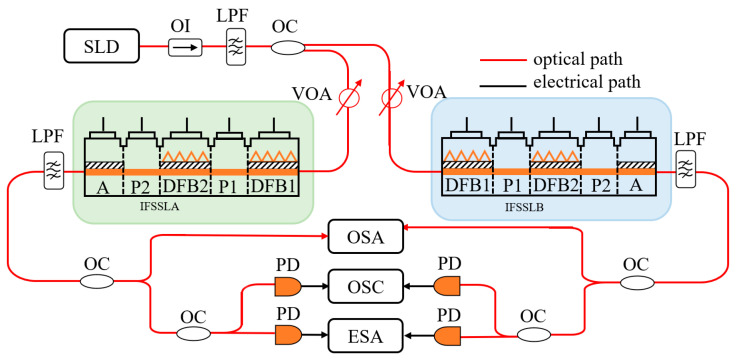
Schematic diagram of chaos synchronization integrated five-section semiconductor lasers. SLD, superluminescent light diode; OI, optical isolator; LPF, low-pass filter; OC, optical coupler; VOA, variable optical attenuator; MSSL, multi-section semiconductor laser; PD, photodetector; m, original message; m’, recovered message.

**Figure 2 entropy-26-00405-f002:**
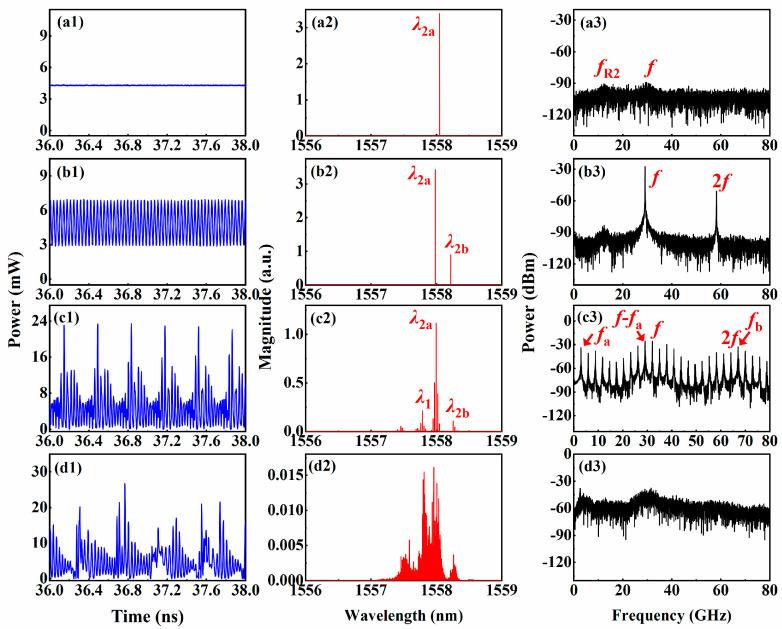
When *I*_P2_ = *I*_A_ = 0 mA, *I*_DFB2_ = 60 mA and *I*_DFB2_ = 66 mA, the temporal waveform (first column), optical spectra (second column) and R-F spectrum (third column) of IFSSL under different *I*_P1_. (**a1**–**a3**) *I*_P1_ = 9 mA; (**b1**–**b3**) *I*_P1_ = 12 mA; (**c1**–**c3**) *I*_P1_ = 15 mA; (**d1**–**d3**) *I*_P1_ = 17 mA.

**Figure 3 entropy-26-00405-f003:**
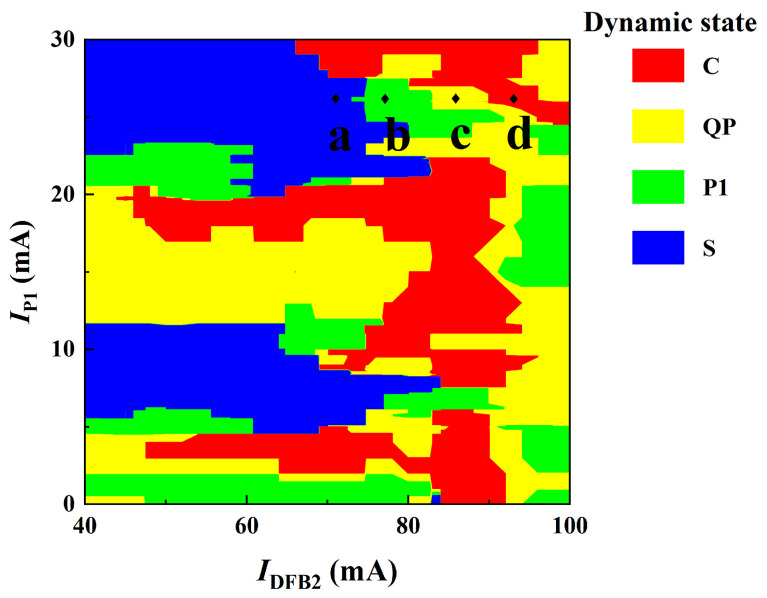
Diagram of dynamic state of the IFSSL in the parameter space of *I*_P1_ and *I*_DFB2_, where *I*_DFB1_ = 60 mA, *I*_P2_ = 0 mA, *I*_A_ = 0 mA and the different colors denote different states.

**Figure 4 entropy-26-00405-f004:**
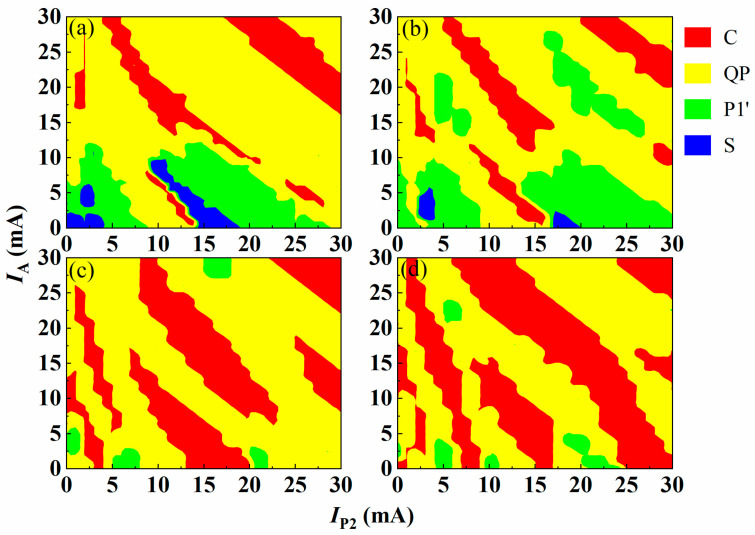
Diagram of dynamic state of the IFSSL under different *I*_P2_ and *I*_A_, When *I*_DFB1_ = 60 mA, *I*_P1_ = 26 mA and *I*_DFB2_ are (**a**) 71 mA, (**b**) 77 mA, (**c**) 86 mA and (**d**) 93 mA. The different colors correspond to different states.

**Figure 5 entropy-26-00405-f005:**
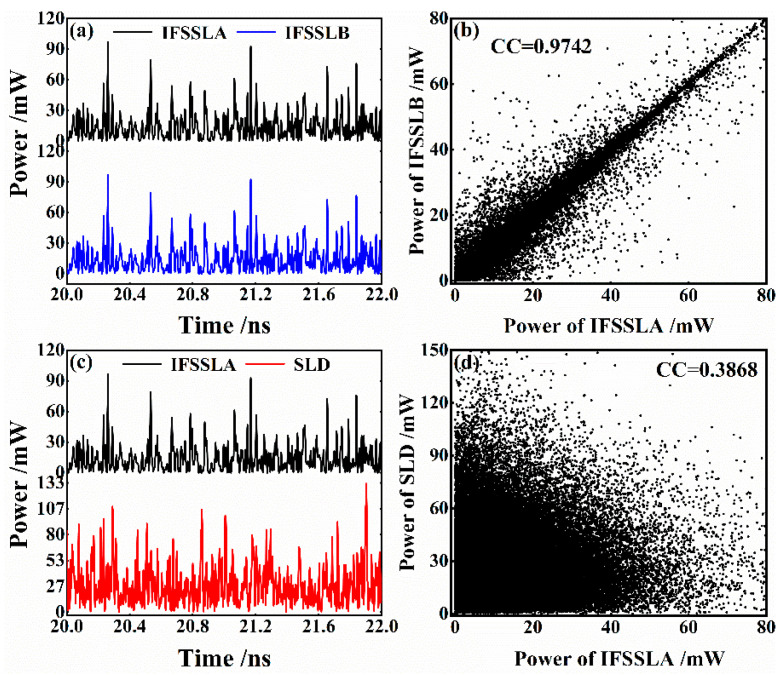
Chaos synchronization of IFSSLA and IFSSLB. (**a**) Temporal waveforms of response IFSSLA and IFSSLB; (**b**) correlation plots of response IFSSLA and IFSSLB; (**c**) temporal waveforms of driving signal SLD and response IFSSLA; (**d**) correlation plots of driving signal SLD and response IFSSLA.

**Figure 6 entropy-26-00405-f006:**
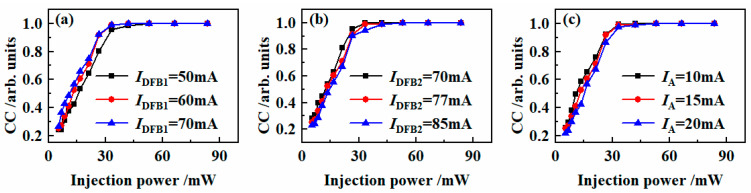
The effects of the injection power on synchronization quality under different conditions (**a**) *I*_DFB1_, (**b**) *I*_DFB2_ and (**c**) *I*_A_ where: *I*_P1_ = 26 mA, *I*_P2_ = 15 mA, (**a**) *I*_DFB2_ = 77 mA, *I*_A_ = 15 mA; (**b**) *I*_DFB1_ = 60 mA, *I*_A_ = 15 mA; (**c**) *I*_DFB1_ = 60 mA, *I*_DFB2_ = 77 mA.

**Figure 7 entropy-26-00405-f007:**
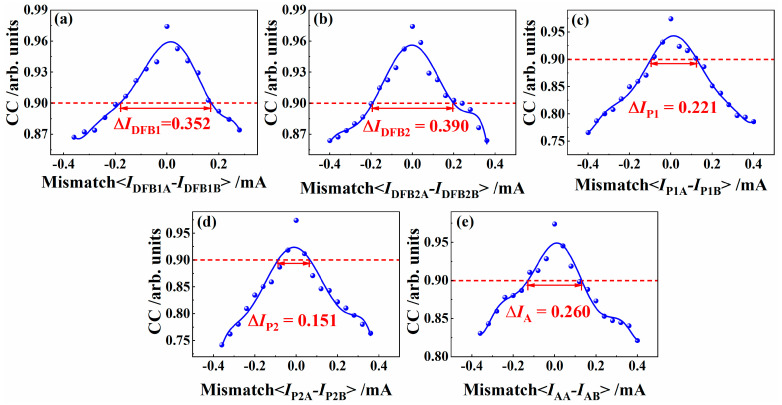
Effects of the current mismatch of the five sections on the synchronization coefficient of IFSSLA and IFSSLB.

**Table 1 entropy-26-00405-t001:** Parameter values of IFSSL used in VPI.

Parameter	Description	Value
*L* _DFB1_	DFB1 section length	200 μm
*L* _P1_	P1 section length	400 μm
*L* _DFB2_	DFB2 section length	200 μm
*L* _P2_	P2 section length	400 μm
*L* _A_	A section length	320 μm
*W* _A_	Active region width	2.50 μm
*a* _I_	Internal loss factor	1000 m^−1^
*α*	Linewidth enhancement factor	4
*C* _i_	Index grating coupling coefficient	15,500 m^−1^
*C* _g_	Gain grating coupling coefficient	2000 m^−1^
*N* _0_	Transparency carrier density	1.40 × 10^24^ m^−3^
*G* _non_	Nonlinear gain coefficient	1.00 × 10^−23^ m^3^
*G* _m_	Linear material gain coefficient	3.00 × 10^−20^ m^2^
*C* _MQW_	Confinement factor MQW	0.07
Λ_g_	Group index	3.70

## Data Availability

The original contributions presented in the study are included in the article material, further inquiries can be directed to the corresponding authors.

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
