# Peer review of "Chaos Synchronization of Integrated Five-Section Semiconductor Lasers"

_entropy, 2024, doi:10.3390/e26050405_

Round 1

Reviewer 1 Report

Comments and Suggestions for Authors

=== SEE THE ATTACHED FILE ===

Comments on the Quality of English Language

It is fine.

Reviewer 2 Report

Comments and Suggestions for Authors

The paper is badly written and suffers from the lack of scientific content.

The current paper looks like a  "VPIcomponentMaker simulation software" tutorial and it is not enough to be published in the scientific journal.

The phenomenon of chaos induced by non-linearity or by external noise, synchronization, and chaotic synchronization, the obtained results are not discussed in the present text.

Reviewer 3 Report

Comments and Suggestions for Authors

In this manuscript, the authors propose and simulate a scheme for chaos synchronization based on five-section semiconductor lasers with matched parameters, for potential application in secure communications. Starting from the dynamical characterization of the proposed configuration, parameter regions of stable, periodic, quasi-periodic and chaotic dynamics are identified. The latter are then used for chaos synchronization operation, which is analyzed in terms of signal correlations and sensitivity to current mismatch between the two lasers. The topic of the manuscript is interesting, the methods utilized by the authors and the results are clearly presented. I suggest publication of this manuscript in Entropy.

Author Response

Thank you very much for taking the time to review this manuscript. We are grateful for your positive feedback , which have undoubtedly improved the quality of our work. We have carefully revised the manuscript. Your time and effort are greatly appreciated.

Round 2

Reviewer 2 Report

Comments and Suggestions for Authors

Publish in the present form

Comments on the Quality of English Language

Publish in the present form